# ProxyFusion: Face Feature Aggregation Through Sparse Experts

**Bhavin Jawade**
University at Buffalo
bhavinja@buffalo.edu

Alexander Stone
University at Buffalo
awstone@buffalo.edu

Deen Dayal Mohan
University at Buffalo
dmohan@buffalo.edu

Xiao Wang
University at Buffalo
xwang277@buffalo.edu

Srirangaraj Setlur
University at Buffalo
setlur@buffalo.edu

Venu Govindaraju
University at Buffalo
govind@buffalo.edu

## Abstract

Face feature fusion is indispensable for robust face recognition, particularly in scenarios involving long-range, low-resolution media (unconstrained environments) where not all frames or features are equally informative. Existing methods often rely on large intermediate feature maps or face metadata information, making them incompatible with legacy biometric template databases that store pre-computed features. Additionally, real-time inference and generalization to large probe sets remains challenging. To address these limitations, we introduce a linear time $\mathcal{O}(N)$ proxy based sparse expert selection and pooling approach for context driven feature-set attention. Our approach is order invariant on the feature-set, generalizes to large sets, is compatible with legacy template stores, and utilizes significantly less parameters making it suitable real-time inference and edge use-cases. Through qualitative experiments, we demonstrate that ProxyFusion learns discriminative information for importance weighting of face features without relying on intermediate features. Quantitative evaluations on challenging low-resolution face verification datasets such as IARPA BTS3.1 and DroneSURF show the superiority of ProxyFusion in unconstrained long-range face recognition setting. Our code and pretrained models are available at: `https://github.com/bhavinjawade/ProxyFusion`

## 1 Introduction

Face recognition (FR) involves generating representations or templates from face images for 1:1 verification and 1:N identification between query media, known as the *probe*, and an enrolled biometric template, known as the *gallery*. Numerous studies have explored novel feature extraction architectures [10] and metric learning-based loss formulations [3, 17] for learning discriminative representations. But, feature extraction is typically the first step in building a face matching system; the second step involves creating robust templates using sets of images or videos of the same individual. This requires fusing or aggregating representations from different face images (or frames, in the case of videos) to obtain a unified template.

Recently, there has been heightened interest in recognizing individuals under extremely challenging conditions, such as from long distances and high altitudes, exemplified by the IARPA BRIAR program. These scenarios introduce novel challenges, making face feature aggregation even more crucial since only a limited set of frames would contain discriminative identity information. In cross-distribution setting, such as low-resolution to high-resolution face matching where a significant distribution gap

38th Conference on Neural Information Processing Systems (NeurIPS 2024).

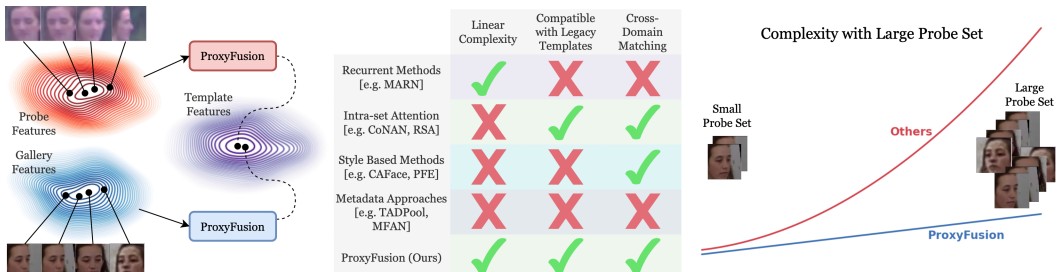

Figure 1: We design our approach to solve three primary challenges (i) Cross-Domain Matching: Matching low-resolution, long-range faces with high-quality gallery faces. (ii) Linear Runtime Complexity: Ensuring our method's time complexity increases linearly with the number of features. (iii) Compatibility with Legacy Templates: Relying solely on final feature vectors for fusion to maintain compatibility with pre-enrolled feature stores that lack intermediate features or metadata.

exists, face feature fusion becomes essential for selecting the most relevant gallery or probe frames for accurate matching.

Existing face feature fusion methods face several key challenges: (i) generalization to a large probe set [9], (ii) real-time inference with low computational cost, (iii) compatibility with legacy template stores, and (iv) cross-distribution matching capabilities [5]. Additionally, these methods should perform importance attribution based on feature quality and feature frequency. This work aims to train a fusion model that learns an order and length-invariant feature weighting strategy. Specifically, given an unordered set of $N$ features, the model should return a set of scores that, when used to weigh the original features, produce an aggregated vector that robustly represents the feature set while maximizing identity-specific information.

As observed by CAFace [9], typical set-to-set attention mechanisms such as Multihead Attention (MHA) and other intra-set attention methods like RSA [12] or CoNAN [5] exhibit quadratic time complexity, $\mathcal{O}(N^2)$. This makes them unsuitable for feature aggregation when dealing with large probe sets. Furthermore, methods such as CAFace [9] require high-dimensional intermediate feature maps ($H \times W \times C$) from the face feature extractor to compute the style information for feature importance weighting. Other approaches, such as [14, 13], leverage external metadata predictors to extract facial characteristics like pose, gender, and distance, which are then used for attribute-conditioned aggregation. These methods are incompatible with existing biometric template stores which typically only store the penultimate features.

In this work we propose a novel feature aggregation framework, ProxyFusion (PF) that utilizes $K$ learnable proxies $\mathbf{P}$, (where $\mathbf{P}_i \in \mathbb{R}^{K \times D}$) to implicitly represent latent facial attributes. These learnable proxies are used for selecting pooling experts based on relevancy scores. Inspired by works on mixture-of-experts [4] and its sparse variants [16], we utilize only the most relevant $\widehat{K}$ experts for each feature-set, thereby reducing the inference time parameters. The proposed pooling experts generate set-centers conditioned on the feature-set distribution. Divergence of input set features from set-centers represents their informativeness. We utilize the divergence to pool the $N$ feature vectors into one aggregated representation.

As we will discuss later, ProxyFusion approach performs order-invariant, size-agnostic feature aggregation without relying on high-dimensional intermediate feature maps or additional metadata. Through various qualitative experiments, we demonstrate that experts can learn distinctive face quality information. Since the effectiveness of face feature fusion is particularly evident in long-range, low-resolution settings, we conduct extensive experiments on the IARPA's BRIAR BTS3.1 dataset [1], which includes videos and images collected in extreme unconstrained environments, along with their constrained counterparts. In addition to BTS3.1, we also conducted experiments on another unconstrained UAV cross-distribution dataset, DroneSURF [6].

In summary, the key contributions of this paper are:

- We introduce ProxyFusion, a feature aggregation framework using learnable query embeddings to select pooling experts. The methods avoids the need for high-dimensional

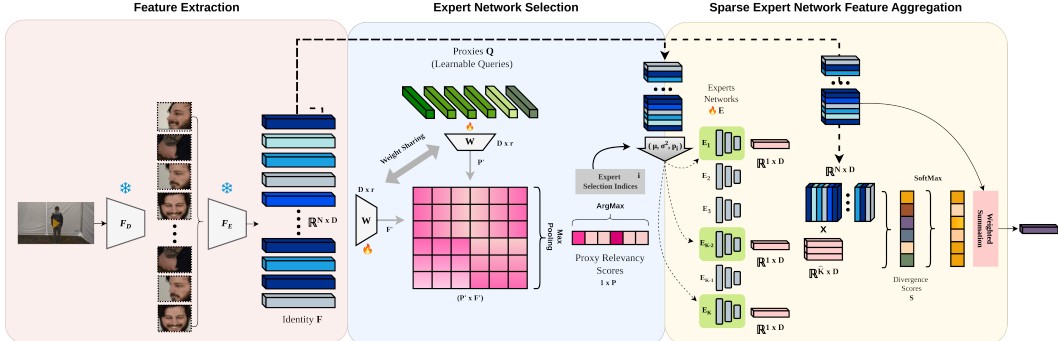

**Feature Extraction**   **Expert Network Selection**   **Sparse Expert Network Feature Aggregation**

Figure 2: An overview of our proposed ProxyFusion Approach. Post feature extraction, our method is divided two end-to-end trainable stages: (i) Expert Selection and (ii) Sparse Expert Network Feature Aggregation. The Expert Selection module takes the $\{\mathbf{f}_i\}_{i=1}^{N}$ and returns the indices of expert networks based on proxy relevancy scores. Next, the selected expert networks compute set-centers conditioned on distribution and aligned proxy. These set-centers attend over the input feature set $\{\mathbf{f}_i\}_{i=1}^{N}$ to compute aggregation weights.

- intermediate feature maps or additional metadata making it compatible with existing biometric template stores.
- By selecting the top experts for each feature set inspired by the mixture-of-experts approach, ProxyFusion significantly reduces inference time parameters.
- Extensive qualitative and quantitative experiments on challenging datasets (IARPA BRIAR BTS3.1 and DroneSURF) show ProxyFusion's effectiveness in long-range, low-resolution face matching, improving FR performance in extreme environments.

## 2 Method

The objective of our method is to correctly match the subject identities in probe media to the subject identities in gallery media. Our problem setting deviates from more conventional face recognition because we have additional challenges where the probe images can have significantly degraded quality as they exist in the long-range, low-resolution domain, where as the gallery images are in the close-range, high-resolution domain. Additionally, the probe feature set for a given subject can be extremely large, possibly in the hundreds of thousands. We define these two sets of images as $\mathcal{I}_G = \{G_1, G_2, \ldots, G_m\}$ for the set of high quality gallery images, and $\mathcal{I}_P = \{P_1, P_2, \ldots, P_n\}$ for the set of low quality probe images. Since a face feature set is composed of faces in different poses and quality, they have different degrees of discriminative identity information. The goal of aggregating features across the feature set $\{\mathbf{f}_i\}_{i=1}^{N}$ is to form a template vector $\mathbf{t} \in \mathbb{R}^{\widehat{K} \cdot d}$ that best represents the subject's identity for cross-distribution matching.

Figure 2 illustrates our proposed approach. Since our method strictly focuses on feature fusion of output embeddings, we pre-extract face features using frozen pretrained face detection and recognition backbones. The face detector is denoted as $f_D : \mathcal{I} \to \mathcal{D}$, where $\mathcal{I}$ is a set of frames and $\mathcal{D}$ is the set of detected face regions. For our face feature extractor, we denote the model as $f_E : \mathcal{D} \to \mathcal{E}$, which maps a cropped face region in $\mathcal{D}$ to a $d$-dimensional feature embedding in the set $\mathcal{E} = \{\mathbf{f}_i\}_{i=1}^{N}$, where $f$ is a face feature vector.

Our proposed feature fusion method is composed of two key parts. The first involves sparsely selecting relevant experts, and the second performs feature pooling using the chosen experts. In section 2.1 we discuss the Expert Network Selection strategy followed by the feature aggregation apporach using these experts.

### 2.1 Expert Network Selection

Given a set $\mathcal{E}$ of face features $\{\mathbf{f}_i\}_{i=1}^{N}$, we define a set of learnable proxies $\{\mathbf{p}_j\}_{j=1}^{K}$, where $\mathbf{f}_i$ , $\mathbf{p}_j$ $\in \mathbb{R}^d$. Here proxies are fixed dimensional embeddings that would represent latent information about facial characteristics required to decide which expert network should be utilized. We project each feature $\mathbf{f}_i$ and proxy $\mathbf{p}_j$ to a unified latent space using a shared projection layer, represented by the

matrix $\mathbf{W} \in \mathbb{R}^{d \times d'}$:

$$\mathbf{f}'_i = \mathbf{W} \cdot \mathbf{f}_i \in \mathbb{R}^{d'}, \quad \mathbf{p}'_j = \mathbf{W} \cdot \mathbf{p}_j \in \mathbb{R}^{d'}$$

where $d'$ is the output dimensionality of the $\mathbf{W}$. We choose $d' << d$ for parameter efficiency.

Using $\mathbf{f}'_i$, $\mathbf{p}'_j$, we compute the **Proxy Relevancy Scores** denoted by $r_j \in \mathbb{R}$, by computing the similarities between each projected proxy $\mathbf{p}'_j$ and all the projected features $\mathbf{f}'_i$, and accumulating these similarities across all features $r_j = \sum_{i=1}^{N} \left( \mathbf{p}'_j \cdot \mathbf{f}'_i \right)$.

To perform expert network selection, we index the top-$\widehat{K}$ values from the set of proxy relevancy scores $\{r_j\}_{j=1}^{K}$. We denote these indices as $\mathcal{I}_{\text{top-}k} = \{j_1, j_2, \ldots, j_k\}$, and use them to selectively activate a subset of expert networks $\{\widehat{\mathbf{E}}_j\}_{j=1}^{\widehat{K}} \subseteq \{\mathbf{E}_j\}_{j=1}^{K}$, where $\widehat{K} < K$. These selected expert networks will be used in the following stage for feature pooling.

## 2.2 Sparse Expert Network Feature Aggregation

We subsampled proxies and their associated expert networks using their relevancy scores with respect to the feature-set in the previous stage. Here we describe our approach to use these sparsely selected expert networks to extract conditional set-centers for aggregating features into the final template vector representation. Motivated by mixture-of-experts [4], we aim to learn mutually exclusive yet homogeneous experts that rank features differently based on learned implicit characteristics. We condition our expert networks over the learned subsampled proxies along with the cross-sample mean and variance across all dimensions within the feature set. More concretely, given a set of feature vectors $\{\mathbf{f}_i\}_{i=1}^{N}$, where each vector $\mathbf{f}_i \in \mathbb{R}^d$, we compute the mean and variance vectors $\boldsymbol{\mu} \in \mathbb{R}^d$, $\boldsymbol{\sigma}^2 \in \mathbb{R}^d$ as follows:

$$\boldsymbol{\mu} = \frac{1}{N} \sum_{i=1}^{N} \mathbf{f}_i, \quad \boldsymbol{\sigma}^2 = \frac{1}{N} \sum_{i=1}^{N} (\mathbf{f}_i - \boldsymbol{\mu})^2.$$

then, we create our input feature distribution representation as $\mathbf{x}_j = \left[ \boldsymbol{\mu} \bigoplus \boldsymbol{\sigma}^2 \bigoplus \mathbf{p}_j \right]$, where $\mathbf{x}_j \in \mathbb{R}^{3 \cdot d}$ and $\bigoplus$ denotes concatenation. For each $\mathbf{x}_j$, we infer through its corresponding expert network, determined by the associated proxy, to obtain the set-centers $\{\mathbf{c}_j\}_{j=1}^{\widehat{K}}$, where $\mathbf{c}_j = \widehat{\mathbf{E}}_j(\mathbf{x}_j), \forall j \in \{1, \ldots, \widehat{K}\}$.

The outputs of the expert networks, referred here as set-centers, are used to compute the divergence of each feature in the original set. These divergence scores are then utilized to compute feature importance. We compute the divergence scores as the un-normalized alignment between the features $\{\mathbf{f}_i\}_{i=1}^{N}$ and set-centers $\{\mathbf{c}_j\}_{j=1}^{\widehat{K}}$ given by $\mathbf{c}_j \cdot \mathbf{f}_i$. Next, we softmax these divergence scores to compute weights as follows:

$$a_{ij} = \frac{\exp(\mathbf{c}_j \cdot \mathbf{f}_i)}{\sum_{k=1}^{N} \exp(\mathbf{c}_j \cdot \mathbf{f}_k)}, \quad \forall i \in \{1, \ldots, N\}, \forall j \in \{1, \ldots, \widehat{K}\}$$

Finally, we compute the weighted sum of the feature vectors for each set-center using the weights $a_{ij}$:

$$\mathbf{s}_j = \sum_{i=1}^{N} a_{ij} \mathbf{f}_i, \quad \forall j \in \{1, \ldots, \widehat{K}\}$$

where $\{\mathbf{s}_j\}_{j=1}^{\widehat{K}}$ is the set of reduced aggregated representation from the selected experts. We define the final template vector $\mathbf{t}$ as concatenation of $\mathbf{s}_j$ given by: $\mathbf{t} = \left[ \mathbf{s}_1, \mathbf{s}_2, \ldots, \mathbf{s}_{\widehat{K}} \right]$.

**Set Length and Order Invariance:** Feature set length invariance refers to our method's ability to effectively handle feature sets of any size, while feature order invariance indicates that our method can process any permutation of features without affecting the outcome. Feature set length invariance arises through the application of mean and variance as a representation of feature-set distribution as also utilized by [5]. Mean and variance could generalize to varying feature lengths if the training-set consists of diversely sampled features. We facilitate this through our batch creation strategy.

To construct a batch, we select a set of $M$ subject identities $\{I_j\}_{j=1}^{M}$ and subsample their respective probe and gallery feature sets. Let $\mathbf{P}_i$ denote the probe feature set for identity $I_i$ and $\mathbf{G}_i$ denote the respective gallery feature set. We define the subsets $\widehat{\mathbf{P}}_i \subseteq \mathbf{P}_i$ and $\widehat{\mathbf{G}}_i \subseteq \mathbf{G}_i$, where the sizes of these

subsets are uniformly sampled from the range $|\widehat{\mathbf{P}}_\mathbf{i}|, |\widehat{\mathbf{G}}_\mathbf{i}| \sim \mathcal{U}(L, U)$. For efficient computation, we zero-pad to the maximum feature set length for any identity within a batch to create uniformly sized tensors.

The order independence of features is a direct consequence of the properties of mean and variance. This invariance also applies to our **Proxy Relevancy Scores**, $\{r_j\}_{j=1}^K$, because the order in which the feature set is multiplied with the proxies is not relevant. The sum is computed along the dimension of the feature vectors, identifying the most relevant proxy without being affected by feature order.

## 2.3   Optimization

We primarily optimize our aggregation experts and the proxies with the identification objective. The identification loss addresses the primary goal to bridge the distribution gap between the probe and gallery templates for a same identity while increasing the inter-class variance across different identities.

The output of *ProxyFusion* is the template $\mathbf{t}$. For all feature subsets in a batch we compute the identity loss using the supervised contrastive loss [7]. Let, $\mathcal{B} = (\mathbf{t}^P, \mathbf{t}^G)$ be the batch consisting of all probe and gallery features then $\mathcal{L}_\text{id}$ is defined as:

$$\mathcal{L}_\text{id} = \sum_{i \in \mathcal{B}} \frac{-1}{|\mathcal{P}(i)|} \sum_{p \in \mathcal{P}(i)} \ln \frac{\exp(\mathbf{t}_i \cdot \mathbf{t}_p^\top / \tau)}{\sum_{j \in \mathcal{A}(i)} \exp(\mathbf{t}_i \cdot \mathbf{t}_j / \tau)}$$

where $\mathbf{t}_i$ is the feature embedding at index $i$ in the set the $(\mathbf{t}^P, \mathbf{t}^G)$. $\mathcal{P}(i)$ is the set of indices of samples with the same subject label as $i$ and $\mathcal{A}(i)$ is the set of indices of all samples with identities different from subject at $i$. Here, $\tau$ is the softmax's temperature parameter.

During training we want to supervise the experts to learn focus on mutually exclusive information through the proxy conditioning. Additionally, while computing the proxy relevancy scores, we want different proxies to attend over different facial characteristics. To achieve this, we propose an additional optimization criteria referred here as the proxy loss. We will fix $K$ uniformly spaced equidistant vectors on the unit hypersphere in $\mathbb{R}^{K-1}$. We choose $K - 1$ dimensions because it is the lowest dimensional space such that our $K$ proxies can be equidistant yet farthest apart from each other. We project the proxies $\mathbf{p} \in \mathbb{R}^d$ down using $\mathbf{W}\mathbf{p} \in \mathbb{R}^{K-1}$. Then, we fix $K$ vectors $\{\mathbf{v}_i\}_{i=1}^K$ to be equidistant from each other as follows:

$$\mathbf{v}_i = \left(\mathbf{e}_i - \frac{1}{d} \sum_{j=1}^d \mathbf{e}_j\right)\sqrt{\frac{d}{d-1}}, \quad \forall d \in \{1, \dots, K-1\}, \forall i \in \{1, \dots, K\}$$

where $\mathbf{e}_i \in \mathbb{R}^{K-1}$ are the standard basis vectors. Based on this the proxy loss is defined as:

$$L_\text{Proxy} = \frac{1}{K} \sum_{i=1}^K \left[ \ln\left(1 + \exp\left(-\alpha(s_{ii} - \lambda)\right)\right) + \frac{1}{|K-1|} \sum_{\substack{k \in K \\ k \neq i}} \ln\left(1 + \exp\left(\beta(s_{ik} - \lambda)\right)\right) \right]$$

Where, $s_{ik} = (\mathbf{p}_i \cdot \mathbf{v}_k) / (\|\mathbf{p}_i\|\|\mathbf{v}_k\|)$ is the similarity between the $i^\text{th}$ proxy and $k^\text{th}$ fixed basis vector. $\lambda$ is the threshold for the exponential function. The first part of this term enforces that proxies get closer to their respective basis vectors, while second part enforces that they move away from all other negative basis vectors.

The final loss is given by $\mathcal{L} = \mathcal{L}_\text{ID} + \gamma \cdot \mathcal{L}_\text{Proxy}$, where $\gamma$ is weightage of the proxy loss.

## 3   Experiments

### 3.1   Dataset

To illustrate the effectiveness of our aggregation technique in long-range, low-resolution, unconstrained matching scenarios, we selected challenging datasets featuring low-quality images and

videos captured from long distances. Our experiments utilize the following datasets for training: (i) **BRIAR Research Set 3 (BRS 3)[1]:** This dataset is from IARPA's BRIAR program Phase 1, featuring videos and images from 170 participants in controlled and field settings. Controlled settings have high-resolution facial images at close range, while field settings include media captured from 100 to 500 meters away. For training CoNAN, we used 49,429 video clips and images from BRS 3, with 20,780 field-setting clips, 23,489 controlled-setting clips, and 5,160 images. Our method is trained on BRS 3 for fair comparison with methods like [5] trained on the same dataset. (ii) **WebFace 4M [22]:** Apart from BRIAR, we also present results by training our method WebFace 4M dataset. This is done to present fair evaluation with CAFace [9] which has been trained on WebFace 4M. Following CAFace [9], we use their randomly sampled subset, consisting of 813, 482 images from 10, 000 identities to train our aggregation function. Following [5], we evaluate our method on following two datasets: (i) **BTS 3.1:** This is the test set for IARPA BRIAR Phase 1 evaluation. We report results for the face-included treatment and control protocols. The treatment set has 5,822 probe videos from 260 subjects in uncontrolled settings, while the control set has 1,914 probe videos from 256 subjects in regulated settings. The BTS 3 gallery is split into Gallery 1 (47,925 clips/images, 485 subjects) and Gallery 2 (47,413 clips/images, 481 subjects), with 351 common distractor identities. (ii) **DroneSURF:** This dataset includes Active and Passive Surveillance settings, each with 100 videos and 786,000+ face annotations. Following [12], we split subjects randomly: 60% (34 identities) for training/validation, 40% (24 identities) for testing. The dataset has 200 videos of 58 subjects, over 411,000 drone-captured frames. Results are based on the video-wise identification protocol.

## 3.2 Implementation Details

**Face Detector and Alignment:** We present results using two popular face detectors: MTCNN [21] and RetinaFace [2]. Unless otherwise specified, the results are reported using the RetinaFace.

**Feature Extractors:** Following previous works [5], [8] we report results using two pretrained frozen feature extractors - Adaface [3] and Arcface [3]. For Adaface [8] we use a ResNet-101 model pretrained on WebFace. Each face image is resized to 112x112x3 before feature extraction. For Arcface, we extract features using a MS1MV2 pre-trained ResNet-50 backbone. For fusion architecture trained using WebFace 4M Adaface features [8], we utilize the precomputed features provided by previous workd [9] for fair comparison.

**Architecture and Hyperparameters:** For the expert networks we utilize a three layer MLP with LeakyReLU activation and a dropout with probability of 0.5. More specifically, the first layer in the MLP projects from 1536 to 1024, the second layer from 1024 to 1024, and last layer from 1024 to 512. Overall the MLP has 3.14M learnable parameters. For supervised contrastive loss, we use a temperature of 0.1. The proxy loss weight $\gamma = 0.01$ and threshold $\lambda = 0.1$. We utilize the Adafactor with adaptive learning rates as the optimizer. For all SoTA experiments we utilize number of total proxies $K$ as 11, and number of selected experts $\widehat{K}$ as 4. We choose the number of identities in a batch $M = 170$ based on available GPU memory. We choose the bounds for probe and gallery subset sizes to be $L = 100$ and $U = 1200$. All experiments are performed on 1 x A6000 48GB NVIDIA GPU. Most experiments require nearly 2 hours with precomputed features.

## 3.3 Discussion and Ablation

**Weight Visualizations and Interpretability:** In Fig. 3, we illustrate the weightings assigned by the set-centers extracted from each selected expert for the face embeddings. This visualization offers insights into the aggregation function's capacity to discern the informativeness of faces. Notably, the model operates solely on frozen precomputed features within the feature set, with no access to intermediate features or metadata.

As evidenced in Fig. 3, each expert attributes higher weights to faces that exhibit more discriminative identity information, such as frontal or profile views, in both the gallery and probe sets. Conversely, the model assigns minimal weight to poor-quality crops, such as images of the back of the head or inaccurate detections. In the case of probes, which are collected in unconstrained, long-range settings, only a limited number of frames in a video contain significant facial information. Hence, the model assigns considerably higher weights to these informative frames within the probe set (e.g., Expert 10 assigns approximately 0.007205 to a frontal face).

In contrast, the gallery set comprises numerous high-quality, informative faces, resulting in the model assigning relatively lower weights to these frames due to their abundance. Consequently, weight

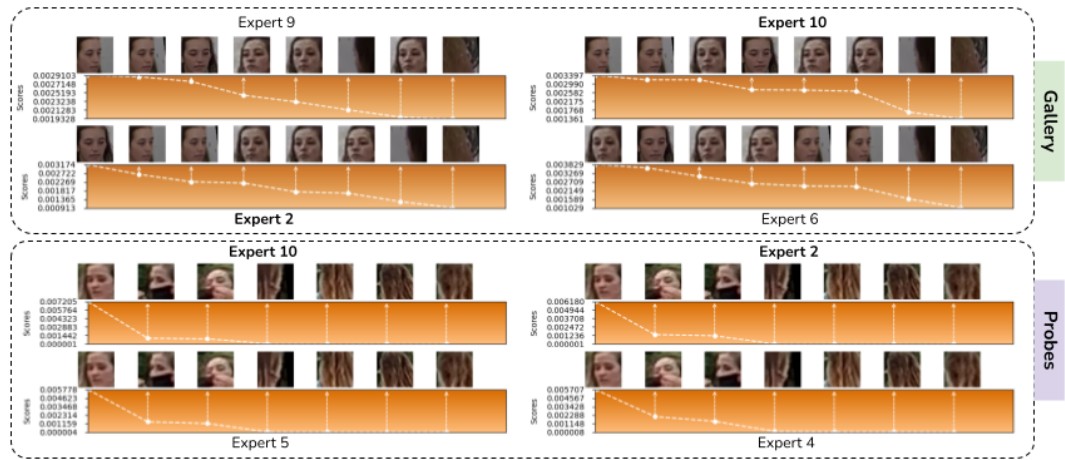

Figure 3: Visualizations of learned weights on BTS3.1 dataset's gallery and probe set. Images on the top are from high quality gallery, and images on the bottom are from low resolution long-range probes. Faces are sorted based on ProxyFusion attention weights from low to high. We present these weights for each of the selected expert.

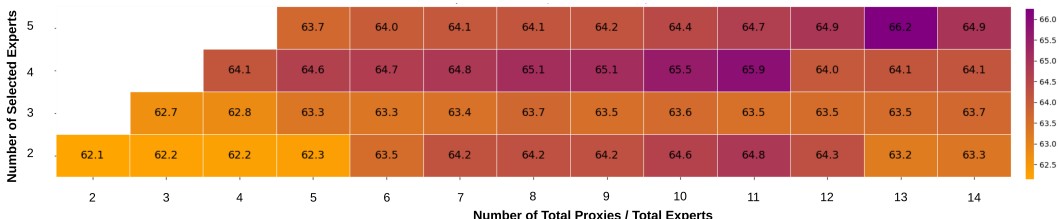

Figure 4: A heatmap of TAR@FAR=$10^{-2}$ on Face Included Treatment Setting of BTS 3.1. The X-axis is the number of selected experts while the Y Axis is total number of experts / proxies.

assignments decrease more gradually in the gallery set compared to the probe set. Additionally, each expert provides subtly different weightings to the same frames, enhancing the overall representation of the face. This analysis demonstrates that the ProxyFusion model adeptly discerns the informativeness of facial features in an interpretable yet effective manner.

**Effect of Number of Proxies:** In Figure 4, we present the face verification performance in terms of TAR@FAR=$10^{-2}$, plotted against the number of proxies $K$ (X-axis) and the selected number of experts $\widehat{K}$ (Y-axis). The results [1] indicate that increasing the number of proxies generally enhances model performance. However, this improvement plateaus around 10-12 proxies, and further increasing the number of proxies beyond 13 leads to overfitting. This overfitting likely occurs because the model, with a higher number of experts during training, becomes more prone to fitting the training distribution and training subjects too closely, thus failing to generalize well to unseen subjects.

**Effect of Number of Selected Experts:** Similarly, Figure 4 also demonstrates that an increase in the number of selected experts generally enhances performance. However, this improvement comes at the expense of longer inference times. As the number of selected experts increases, the number of parameters required during inference also rises, leading to increased

Table 1: Performance analysis of the model while training with and without proxy loss with varying number of selected experts.

|  | $\widehat{K} = 2$ | $\widehat{K} = 3$ | $\widehat{K}=4$ | $\widehat{K}=5$ | $\widehat{K}=6$ |
|---|---|---|---|---|---|
| No $\mathcal{L}_{\text{Proxy}}$ | **62.49** | 64.15 | 67.32 | 67.86 | 66.21 |
| $\mathcal{L}_{\text{Proxy}}$ | 62.10 | **65.37** | **68.93** | **68.57** | **67.03** |

---

[1]It should be noted here that, given the number of experimental runs required to perform this analysis we only indicate the results after $10^{th}$ epoch in this analysis, which may not reflect the best performance of the model.

Table 3: Verification Performance (TAR (%) @FAR=%) for face included treatment and control protocols of the BTS 3.1 dataset. All faces are detected and aligned using RetinaFace face detector.

| | Feature | Dataset | Face Included Treatment | | | | Face Included Control | | | |
|---|---|---|---|---|---|---|---|---|---|---|
| | | | $10^{-1}$ | $10^{-2}$ | $10^{-3}$ | $10^{-4}$ | $10^{-1}$ | $10^{-2}$ | $10^{-3}$ | $10^{-4}$ |
| GAP [11] | Adaface [8] | Briar | 76.6 | 58.4 | 43.3 | 32.1 | 98.5 | 94.6 | 88.9 | 81.2 |
| NAN [20] | Adaface [8] | Briar | 78.5 | 61.2 | 46.8 | 33.4 | 98.5 | 95.3 | 89.3 | 84.8 |
| MCN [19] | Adaface [8] | Briar | 79.4 | 62.9 | 47.3 | 35.9 | 98.5 | 95.9 | 90.7 | 85.7 |
| CoNAN [5] | Adaface [8] | Briar | 81.3 | 64.3 | 49.6 | 36.8 | 98.6 | 96.2 | 91.8 | 86.1 |
| ProxyFusion | Adaface [8] | Briar | **83.7** | **68.9** | **53.9** | **40.1** | **98.6** | **96.8** | **92.7** | **88.3** |

inference time. Our observations indicate that setting the number of selected experts, $\widehat{K}$, to 4 achieves performance close to the best model while significantly reducing computation time.

**Contribution of Proxy Loss $\mathcal{L}_{\text{Proxy}}$:** In Table 1 we present the contribution of the $\mathcal{L}_{\text{Proxy}}$ to the overall performance. We observe that though the contribution of $\mathcal{L}_{\text{Proxy}}$ is marginal when the number of selected experts is small, there are performance improvements at modest $N$. We believe this is due to the decorrelation of proxies, which results in extraction of diverse information from the feature set.

### 3.4   Inference Time and Computational Cost:

To validate our algorithm's claimed linear time complexity we perform GFLOPs analysis of our method against increasing sizes of the feature-set. As can be observed from Fig. 5, we start with small feature-set size of 100 and scale it up to 1 million. Our fusion model's (in blue) GLOPs increases linearly with the increasing number of features in the set. On contrary, GFLOPs increase quadratical for models like CoNAN [5] and [9]. Moreover, these with single-shot input of large feature-sets to these models, the memory footprint increases quadratically due to intra-set attention

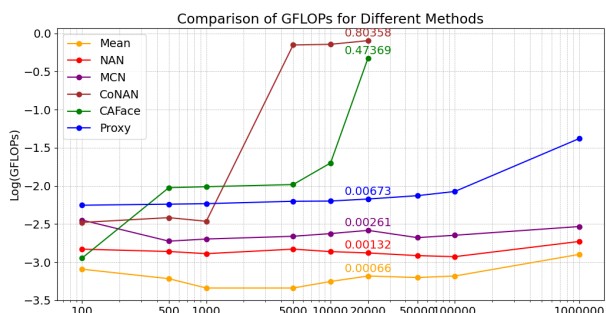

Figure 5: Time complexity comparison of ProxyFusion approach against SoTA. On the Y-axis we plot the Log of GFLOPs with base 10, and X axis is the number of features in the feature set $N$.

which leads to out-of-memory issues for these models beyond an N of 21000. These experiments are performed on 1 NVIDIA A6000 averaged over 3 runs.

### 3.5   Comparison to SotA methods

In table 3 we compare our results with the state-of-the-art (SOTA) in face verifcation using the RetinaFace face detector and Adaface Features on BTS3.1 Dataset. In table 4 we present results using MTCNN face detector and Adaface and Arcface feature extractors on BTS3.1. As mentioned earlier, for fair comparison to methods such as [9], in first half of table 4 we train our model on WebFace dataset using the pre-computed Adaface features released by [9].

As can be observed from table 3 and 4 our method significantly outperforms other linear time methods such as NAN[20] and MCN[19] while providing a significant jump over naive

Table 2: Rank-1 accuracy (%) for video-wise identification on DroneSURF dataset.

| Trained On DroneSURF | | | |
|---|---|---|---|
| | Feature | Active | Passive |
| GAP [11] | Adaface [8] | 46.87 | 7.29 |
| NAN [20] | Adaface [8] | 65.62 | 6.25 |
| MCN [19] | Adaface [8] | 72.92 | 8.33 |
| CoNAN [5] | Adaface [8] | 80.21 | 13.54 |
| **Ours** | **Adaface** [8] | **83.33** | **13.54** |
| Trained On BRS (Cross-Dataset Evaluation) | | | |
| GAP [11] | Adaface [8] | 46.87 | 7.29 |
| NAN [20] | Adaface [8] | 80.21 | 8.33 |
| MCN [19] | Adaface [8] | 79.16 | 10.41 |
| CoNAN [5] | **Adaface [8]** | **83.33** | 12.50 |
| **Ours** | Adaface [8] | 80.21 | **13.54** |

Table 4: Verification Performance (TAR (%) @FAR=%) for face included treatment and control protocols of the BTS 3.1 dataset. All faces are detected and aligned using MTCNN face detector.

| | Feature | Dataset | Time | No Inter. | Face Incl. Trt. | | | Face Incl. Ctrl. | | |
|---|---|---|---|---|---|---|---|---|---|---|
| | | | | | $10^{-2}$ | $10^{-3}$ | $10^{-4}$ | $10^{-2}$ | $10^{-3}$ | $10^{-4}$ |
| GAP [11] | Adaface [8] | WF4M | $\mathcal{O}(N)$ | ✓ | 50.8 | 40.8 | 31.7 | 91.3 | 86.9 | 80.1 |
| CAFace [9] | Adaface [8] | WF4M | $\mathcal{O}(N^2)$ | ✗ | 51.3 | **42.0** | 33.4 | 92.7 | 88.8 | 83.0 |
| ProxyFusion | Adaface [8] | WF4M | $\mathcal{O}(N)$ | ✓ | **51.5** | 41.7 | **33.69** | **92.9** | **89.2** | **83.2** |
| GAP [11] | Arcface [3] | Briar | $\mathcal{O}(N)$ | ✓ | 37.0 | 27.3 | 19.5 | 84.8 | 75.3 | 66.2 |
| NAN [20] | Arcface [3] | Briar | $\mathcal{O}(N)$ | ✓ | 39.0 | 26.6 | 18.3 | 84.3 | 72.9 | 60.4 |
| MCN [19] | Arcface [3] | Briar | $\mathcal{O}(N)$ | ✓ | 39.4 | 28.2 | 19.4 | 87.1 | 77.9 | 67.2 |
| CoNAN [5] | Arcface [3] | Briar | $\mathcal{O}(N^2)$ | ✓ | **43.4** | **32.1** | 23.1 | 87.6 | 81.0 | 71.9 |
| **ProxyFusion** | Arcface [3] | Briar | $\mathcal{O}(N)$ | ✓ | 42.1 | 31.4 | **23.4** | **88.4** | **81.8** | **73.9** |

∗ Inter. denotes whether the method depends on intermediate features for feature fusion. If a method requires intermediate features, it is not compatible with legacy templates.

averaging or Global Average Pooling (GAP) [11]. Our method also achieves on-par performance to quadratic time complexity methods such as CoNAN [5] and CAFace [9]. When trained on Web-Face, our approach outperforms CAFace [9] on most FAR thresholds without utilzing intermediate features unlike [9]. When compared to CoNAN [5], our method performs outperforms it on Face included control setting while being significantly faster (See Fig. 5, having much lower runtime memory requirements, and comparable inference time parameters (13.6M for CoNAN and 12.4M for ProxyFusion). Similar trend can be observed on DroneSURF dataset (refer Table 2). All SotA experiments are performed several times and we present the average results. We calculated Standard Errors of the Mean (SEM) across the multiple observations and found the SEM to very low in the range of 0.008-0.09 showing the statistical significance of the experiments.

## 4 Related Work

**Methods based on Intermediate Feature Maps and Metadata:** Leveraging intermediate feature maps for feature fusion, methods like the nonlocal neural network [18] and RSA [12] show promise in modeling intra-set relationships through detailed spatial analysis. These methods utilize intermediate feature maps $U_i$ of size $C_m \times H \times W$ to capture complementary information and refine spatial relationships. CAFace [9] addresses the computational issue of these approaches with a two-stage feature aggregation method, using these feature maps as style information. This involves assigning $N$ inputs to $M$ global cluster centers and fusing clustered features. CAFace generates an affinity map with $N^2$ complexity [15]. The requirement for intermediate feature maps makes these methods incompatible with legacy biometric systems lacking precomputed intermediate features. In contrast, our method relies solely on the penultimate feature representation, typically used for generating biometric templates. Furthermore, Metadata extracted from face images can serve as an alternative source for inferring face quality. [14] introduced a feature aggregation method that integrates metadata (e.g., yaw, pitch, face size) and utilizes a siamese network to determine the relative quality correlations among face images in a set. Additionally, metadata can facilitate fine-grained matching across galleries and probes by identifying faces with similar characteristics. For instance, TADPool [13] adapted both probe and gallery features by considering attributes like face yaw and roll, employing an attention block to pool probe features based on their compatibility with selected gallery features.

**Linear Time Feature Fusion Methods:** Previous works have proposed aggregation methods with linear time complexity against the number of features in set. In [20], a network architecture with two cascaded attention blocks is presented, which evaluates the significance of features in an image set and uses these scores for feature aggregation. Recently, [15] employed a differentiable coreset selection approach, using learned metrics and a Gumbel-Softmax distribution to optimize the selection of a small, representative subset, which is then enriched via self and cross-attention mechanisms thereby reducing computational complexity by decreasing set-size. [19] introduces a multicolumn network that assigns weights to images within a set based on their visual quality, assessed through a self-quality assessment module. This network then dynamically adjusts these weights according to each image's relative content quality compared to others in the set.

## Conclusion

We address key challenges in feature fusion: (i) real-time inference, (ii) cross-distribution matching, (iii) generalization to large feature sets, and (iv) compatibility with legacy feature stores. Our proxy-relevancy based approach for expert selection, combined with a feature pooling approach, ensures robust multifaceted feature aggregation with minimal inference time and no need for extra metadata or intermediate features. Ablation studies demonstrate our method's superior performance and efficiency, outpacing existing approaches in both speed and effectiveness for real-time applications.

**Potential Negative Societal Impacts:** We are meticulous about training or testing our models only on datasets with approved IRB and consent from involved human subjects. This is why we do not use IJB-B/C datasets, as they contain web-sourced faces without consent or IRB approval. Conversely, datasets like IARPA BRIAR and [6] which we utilize are collected under IRB with subject consent.

**Limitations:** While our method effectively discerns feature informativeness via sparse experts, relying solely on feature-set distribution statistics overlooks fine-grained intra-set relationships. We can enhance this by hierarchically merging them with intra-set methods, achieving further gains with limited computational cost.

## Acknowledgments

This research is based upon work supported in part by the Office of the Director of National Intelligence (ODNI), Intelligence Advanced Research Projects Activity (IARPA), via 2022-21102100001. The views and conclusions contained herein are those of the authors and should not be interpreted as necessarily representing the official policies, either expressed or implied, of ODNI, IARPA, or the U.S. Government. The U.S. Government is authorized to reproduce and distribute reprints for governmental purposes notwithstanding any copyright annotation therein.

This works is also partially supported by the the National AI Research Institutes program by the National Science Foundation (NSF) and the Institute of Education Sciences (IES), U.S. Department of Education, through Award #2229873. Any opinions, findings and conclusions or recommendations expressed in this material are those of the author(s) and do not necessarily reflect the views of the NSF, the IES, or the U.S. Department of Education.

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
