# OpenReview forum: "ProxyFusion: Face Feature Aggregation Through Sparse Experts"
_NeurIPS.cc/2024/Conference — NeurIPS 2024 poster_

### Official Review · Reviewer_Cxak · 2024-07-04

**Soundness:** 3
**Presentation:** 3
**Contribution:** 3
**Rating:** 4
**Confidence:** 5

**Summary:**

This paper presents a novel face recognition framework that aims to address the challenges of feature fusion in long-range, low-resolution scenes. The authors propose a linear time complexity approach that is compatible with traditional biometric template databases and does not require additional metadata or intermediate feature maps.ProxyFusion utilises a set of learnable agents to implicitly represent potential facial attributes and selects the most relevant experts to focus on the feature set, thus significantly reducing inference time and parameters. The method is order and size invariant, making it very robust for real-time applications and large probe sets. Extensive experiments on the IARPA BRIAR BTS3.1 and DroneSURF datasets demonstrate the superiority of ProxyFusion in unconstrained long-range face recognition settings.

**Strengths:**

1. In this paper, the authors propose a novel feature aggregation framework to address key challenges in face recognition, especially in low-resolution and long-range scenes.
2. The runtime of ProxyFusion's feature aggregation method is linearly related to the number of features, which is very effective for real-time reasoning and processing large probe sets.

**Weaknesses:**

1. The optimization of the sparse experts is ambiguous. In the section of methods, the authors do not provide any details on how to optimize the sparse experts.
2. The paper mentions that relying solely on feature-set distribution statistics might overlook fine-grained intra-set relationships. This suggests that the model may not capture all the nuances present within a set of features.
3. The performance of ProxyFusion, like any face recognition system, is likely to depend heavily on the quality of the input data. Poor quality inputs could degrade performance.
4. The paper does not mention the model's robustness to adversarial attacks, which is an important consideration in the field of biometrics and face recognition.

**Questions:**

Please refer to the weaknesses.

**Limitations:**

Please refer to the weaknesses.

---

> ### Author Rebuttal · Authors · 2024-08-07
>
> 1. **Optimization:** Thank you for your feedback. For clarity on the optimization of the sparse experts, please refer to section 2.3 titled "Optimization" in our paper. There, we elaborate on the overall optimization objective. Here's a more detailed explanation of our optimization strategy.
> Our loss has two terms:
>
> > **Identity Loss**: This component ensures that the aggregated embeddings from the same identity—collected under varying qualities—are drawn closer, while distancing them from the embeddings of different identities. We implement this through a supervised contrastive loss.
>
> > **Proxy Loss**: During training, this loss helps ensure that proxies distinctly and exclusively represent specific facial features. This distinctiveness is critical for selecting the appropriate experts for different facial characteristics.
>
> To enforce that proxies represent mutually exclusive facial charachterstics, we initialize a fixed number (K) of equidistant and equiangular standard basis vectors. Each proxy is encouraged to align closely with its corresponding vector and diverge from the others. This is enforced through the following formulation:
>
> $$ L_{\text{Proxy}} = \frac{1}{K} \sum_{i=1}^K \left[ \ln \left( 1 + \exp \left( -\alpha (s_{ii} - \lambda) \right) \right) + \frac{1}{|K-1|} \sum_{k \in K \ \text{and} \ k \neq i} \ln \left( 1 + \exp \left( \beta (s_{ik} - \lambda) \right) \right) \right] $$
>
> In this equation, $\( s_{ik} \)$ denotes the cosine similarity between the $\(i^{th}\)$ proxy and the $\(k^{th}\)$ basis vector. The first term of the equation encourages each proxy to approach its specific basis vector, while the second term ensures separation from the remaining vectors. Parameters $\( \alpha \)$ and $\( \beta \)$ scale the positive and negative similarities, respectively, with $\( \lambda \)$ set at 0.1 as the threshold. This structured approach helps maintain the distinctiveness and efficacy of each proxy throughout training.
>
> 2. Relying on feature-set distribution statistics may not capture fine-grained intra-set relationships as exhaustively as an O(n^2) approach such as attention. However, our method aims to approximate these relationships in O(n) through feature-set distribution statistics. We believe that our method accurately approximates these relationships based on our method’s superior performance to $O(n^2)$ methods such as CAFace [8] and CoNAN [5].
>
> 3. Thank you for your comment. Our method is specifically designed to excel in long-range and low-resolution data scenarios, addressing the challenges posed by such poor-quality inputs. Our method's superior performance in this poor-quality regime as evident from the performance on the Face Included Treatment setting of BTS3.1, where face images are uncontrolled and low quality.
>
> 4. While we agree that this is an important aspect of biometrics and face recognition, our paper solely focuses on improving face feature fusion performance under challenging low-resolution and uncontrolled conditions and leaves adversarial robustness for future works. We would like to emphasize that adversarial study of face recognition methods is not the objective of this paper.

---

### Official Review · Reviewer_BueQ · 2024-07-05

**Soundness:** 3
**Presentation:** 2
**Contribution:** 3
**Rating:** 5
**Confidence:** 3

**Summary:**

The paper proposes a novel feature fusion technique for unconstrained face recognition via sparse experts. The authors propose a expert network selection mechanism using the $k$ proxies, $p_j$, and all the $N$ precomputed face features, $f_i$. The top $\hat{k}$ selected experts are used by the 3d set centers to produce discriminative features per expert that finally forms the template. Two losses are proposed, a supervised contrastive loss ($L_{id}$) to ensure high interclass and low intra class separability between probe and gallery, and a proxy loss ($L_{proxy}$) to ensure the experts learn mutually exclusive information. The authors validate their method primarily on DroneSurf and BRIAR dataset on different TAR@FAR=thresholds. They also provide visualization for the learned experts and GFLOPS analysis to support the linear time complexity claim.

**Strengths:**

* The application of mixture of experts acts as a way of diffentiable sampling to select the right experts and intrinsically, the right feature for aggregation.
* The paper presents a strong evidence emperically, especially with the BRIAR dataset, which is a highly noisy and a challenging dataset.
* The authors detail the contribution on all aspects that includes - interpretability of weights for set-centers, effects of number of proxies, effects of expert selection, and finally, the proxy loss.

**Weaknesses:**

### 1. The presentation of the method end experiment section could be greatly improved. There are some texts that is unclear that certainly affects the final rating of the paper.

a.  Line 97, What are proxies in this context and how are they constructed initially? Are they randomly initialized? Or are they some form of router network? The paper also seems to use proxies/learnable queries (line 68) interchangeably which is further confusing.

b.  Line 125, Divergences represent the dissimilarity between two distributions. Can the authors please expand on how the dot product between two feature vector is a divergence?

c.  Table 1, What dataset has this been tested on? Is this where the authors employed $\hat{K}=4$ as the optimum experts?

d.  What is the success rate of both face detectors for probe images? At 500m with several noise factors, these standard detectors might encounter higher degree of failure.

e.  Figure 4 is a bit confusing as to what exactly $x-axis$ is, as the experts/proxies are used interchangeably.

f.  It would also add value if authors are able to add TAR@FAR values per each distance category for probe BTS 3.1. I am assuming this would involved categorizing the results which should not be a completely new experiment.


### 2. The paper does not include enough description to make it reproducible.

a.  Line  141, what were the chosen $L$ and $U$ values in the experiments?

b.  Line 143, what was the suitable $M$ found to be?

c.  Line 146, pad them with what value?

d.  What is the final $d$ and $d'$ ?


### 3. I also have few minor corrections to suggest to further improve the quality of the paper.

a.  Paragraph 139 and 143 seems to convey the same meaning. Can be combined into one.

b. SCL equation for $L_{id}$ uses the transpose $T$ inconsistently.

c. Line 187, I think it is supposed to be "For training *ProxyFusion*..."

d. Other errors:

 * Typo in line 90, *refere*
* Line 125, 'that is used to compute...'
* Line 154, "primary goal bridge...."
* Line 282, "these with single shot..."

**Questions:**

I have mentioned most of the questions in the weakness section. The most important ones to address would be 1 and 2.

**Limitations:**

The authors adequately address the limitations about the proposed methods and use of only IRB approved datasets for experiementation.

For future work, it would be interesting to compare the method against [1], which also attempts to achieve similar goals. This however, is a suggestion, and in no way affects my final rating for this conference.


[1] Shapira, Gil, and Yosi Keller. "FaceCoresetNet: Differentiable Coresets for Face Set Recognition." Proceedings of the AAAI Conference on Artificial Intelligence. Vol. 38. No. 5. 2024.

---

> ### Author Rebuttal · Authors · 2024-08-07
>
> We are grateful for your valuable feedback. We have addressed your questions below.
>
> 1.a) We define proxies as learnable embeddings / vectors of dimension length 512 (same dimensionality as the face-feature embeddings). These embeddings represent latent information about facial characteristics required to decide which expert network should be utilized. We initialize proxies using kaming-he initialization, and then enforce them to be farthest apart using our proposed proxy-loss. Thanks for suggesting to not use learnable queries and proxies interchangeably as in line 68. We will correct this in camera-ready.
>
> 1.b) We compute the similarity of the original input features to the set-centers extracted from the selected expert-networks. These similarity scores tell us how much is an original feature diverging (or dissimilar) from the set-center - smaller the similarity, more diverging/dissimilar is the feature from the set-center. Hence we utilize the term divergence scores to describe these scores. These are then softmax-normalized and utilized as weightages for feature aggregation.
>
> 1.c) Thanks for noticing this. Table 1 results are on BTS3.1 dataset, face-included treatment setting with Adaface feature extractor. We will provide this information explicitly in the camera-ready paper. The optimum value of $\hat{K}$ = 4, can be discerned from Figure 4 and Table 1. Figure 4 shows that at $\hat{K}$=4 number of selected experts and 11 number of total experts, the model reaches closest to best performance. The ablation is present on the test-set, the selection of $\hat{K}$=4 was performed on a subject disjoint validation-set.
>
> 1.d) We utilize standard detectors, specifically MTCNN and Retinaface in this work for fair comparison to previous works such as CAFace [8], CoNAN [5] etc. Performance of the face-detectors at long distances such as 500m is not directly benchmarked in previous works due to lack of human-annotated ground truths. Change of face-detectors significantly impacts the overall recognition performance as evident from Table 2 and Table 3 since better face detectors such as Retinaface detect more number of faces in a video leading to better recognition performance post fusion.
>
> 1.e) Total number of proxies and total number of experts are always equal, since proxies help in selecting the right expert for a given feature-set. X axis on Figure 4 denotes this “total” number of proxies (or total number of experts since they are equal). Y axis in Figure 4 denotes the number of selected top-k experts. This is always less than the total number of experts. In summary Figure 4 presents an ablation on the performance of the model as the total-number of experts and number of selected top-k experts change. We will make this more clearer in the camera-ready.
>
> 1.f) Thanks for the great suggestion. Below is one set of segregated results based on distance. We will provide more detailed analysis in the final-version.
>
> | Distance | Method       | TAR@FAR=10^-1 | TAR@FAR=10^-2 | TAR@FAR=10^-3 | TAR@FAR=10^-4 |
> |----------|--------------|---------------|---------------|---------------|---------------|
> | 500m     | GAP          | 46.72%        | 36.31%        | 29.22%        | 23.75%        |
> | 500m     | **ProxyFusion** | **48.57%**    | **38.94%**    | **31.84%**    | **25.91%**    |
> | 200m     | GAP          | 62.13%        | 51.20%        | 44.53%        | 36.27%        |
> | 200m     | **ProxyFusion** | **62.93%**    | **53.87%**    | **46.67%**    | **37.87%**    |
> | 100m     | GAP          | 69.27%        | 57.80%        | 48.41%        | 39.17%        |
> | 100m     | **ProxyFusion** | **70.70%**    | **60.83%**    | **51.27%**    | **41.72%**    |
> | <10m     | GAP          | 70.74%        | 62.38%        | 55.82%        | 48.05%        |
> | <10m      | **ProxyFusion** | **71.85%**    | **64.79%**    | **58.48%**    | **50.90%**    |
>
> 2.a) Thanks for bringing this to our notice. The chosen value of L is 100 and U is 1200 during training.
>
> 2.b) M denotes the number of identities sampled to create one batch. We utilize M as 170 based on our GPU’s memory capacity.
>
> 2.c) We pad with “zero-vectors” of dimensionality D to match the length of the biggest feature-set in the batch. This is just performed to create uniform shape tensors for training.
>
> 2.d) The value of d is 512, and d’ is 10.
>
> 3.a) Thanks for noticing this. We will combine the paragraphs into one for camera-ready.
>
> 3.b) Thanks for noticing this. We will add transpose to the denominator.
>
> 3.c) Thanks for noticing this. We will make the correction.
>
> 3.d) We will thoroughly amend the paper for typographical and grammatical errors. Thanks for your comments.

---

> > ### Comment · Reviewer_BueQ · 2024-08-12
> > **Read Acknowledgment**
> >
> > I have read the rebuttals from the Authors. I thank them for the thorough explanation for each question.
> >
> > For 1a, After reading through the explanations here and having gone through the code provided, I have a better understanding about proxies. Please ensure the below points are covered or expanded in the paper so that the same level of clarity is reflected in the final version.
> >
> > * Definition of proxies and its purpose at the beginning of 2.1.
> > * The contrastive-like Intuition behind the proxy loss equation.

---

> ### Author Response · Authors · 2024-08-13
> **Thanks for your response**
>
> Thank you for your response. We appreciate the time and the valuable feedback you have provided. We are glad that our explanations and code could provide better clarity. As you have suggested, we will ensure that the points about the definition of proxies, their purpose, and the contrastive-like explanation behind the proxy loss equation are covered in detail in the final version with the same level of clarity.

---

### Official Review · Reviewer_waA1 · 2024-07-09

**Soundness:** 3
**Presentation:** 4
**Contribution:** 2
**Rating:** 5
**Confidence:** 3

**Summary:**

The authors introduce a novel approach for face feature fusion, addressing typical scenarios such as low-quality probes and high-quality or different domain gallery sets of faces. They describe alternative approaches and conclude that these are currently limited because they are either not compatible with legacy biometric databases or have real-time performance issues.

The authors provide a method for feature aggregation through Sparse Experts. The overall pipeline includes three steps: Feature Extraction -> Expert Network Selection -> Sparse Expert Network Feature Aggregation. Each step is described in detail.

Optimization is done using a two-component loss: Loss for the identification objective + proxy loss. The proxy loss aims to make different proxies attend to different facial characteristics. The authors present experimental justifications for the second loss in the experimental section.

The authors describe the experiments conducted. Two datasets were used for training: BRIAR Research Set 3 and WebFace 4M. Evaluation was performed on BTS 3.1 and DroneSURF. The authors provide all details about these steps. They also present visualizations showing that each expert assigns higher weights to faces with more discriminative identity information and minimal weights to poor-quality images. Experimentally, they found that the optimal setup is to use 4 Selected Experts and 12 total Proxies/Experts.

The authors compared their method with various state-of-the-art alternatives and demonstrated that their method either produces strong results, achieving SOTA performance, or performs on par with quadratic-time methods, despite being a linear-time method.

Additionally, the authors justify that their method is feature order and set length invariant.

**Strengths:**

The idea of the method is quite clear. The algorithm has a linear time complexity, while alternatives with similar quality have quadratic time complexity. The method is invariant to set length and feature order. It can be applied even to legacy databases, which is highly important.
The paper is clearly written, and the results seem to be very promising. The visualizations are well drawn, clear, and insightful.

**Weaknesses:**

The authors commit to releasing the code but did not attach it. While the method is clear and straightforward, the overall implementation can be complex, which could hinder reproducibility. Therefore, I believe the code is a crucial part of this work and should be shared along with the article, especially since this work does not involve heavy neural models or long training procedures.

Without a full code review and the ability to run it, there may not be sufficient proof of reproducibility. Additionally, the authors' response to Question 7, 'Experiment Statistical Significance,' may not fully address the issue, as comparison with alternative methods alone is not evidence of statistical significance.

Furthermore:

> (ii) DroneSURF: This dataset includes Active and Passive Surveillance settings, each with 100 videos and 786,000+ face annotations. Following [12], we split subjects randomly: 60% (34 identities) for training/validation, 40% (24 identities) for testing. The dataset has 200 videos of 58 subjects, over 411,000 drone-captured frames. Results are based on the 203 video-wise identification protocol.

It is unclear how the random split can be compared to other methods. If the split was done as in [12], this information should have been provided. Additionally, I suggest that the splits be released so future authors can make clear comparisons.

**Questions:**

1. Could the authors provide the code and necessary weights at this review stage to enhance clarity and reproducibility?
2. Could the authors provide more detailed information about the DroneSURF random splits or share the splits themselves?
3. Could the authors clarify about the statistical significance?

**Limitations:**

There is one limitation described:

> While our method effectively discerns feature informativeness via sparse experts, relying solely on feature-set distribution statistics overlooks fine-grained intra-set relationships.

---

> ### Author Rebuttal · Authors · 2024-08-07
>
> **Response to Reviewer Comments**
>
> We appreciate the thoughtful critique and the opportunity to enhance the clarity and reproducibility of our work. Below are our responses to the concerns raised:
>
> 1. **Code Availability and Implementation Details:**
>    Thank you for emphasizing the importance of code availability for reproducibility. We have now made the code publicly available on an anonymous GitHub repository to facilitate the review process. It can be accessed at [https://github.com/anonymousdoubleblindreview/ProxyFusion](https://github.com/anonymousdoubleblindreview/ProxyFusion), complete with instructions for installing the necessary requirements. Please note that the datasets used for training and evaluation are externally sourced, and access requests should be directed to the original authors of those datasets as detailed in our repository links.
>
> 2. **Statistical Significance of Experiments:**
>    We acknowledge the need for a rigorous demonstration of statistical significance and have addressed this issue as follows:
>    - **Error Margins:** We conducted our experiments ten times, presenting results alongside their corresponding Standard Errors of the Mean (SEM), calculated as
> $$
> \(\text{SEM} = \frac{s}{\sqrt{n}}\)
> $$
> , where $\(s\)$ is the standard deviation and $n$ is number of observations. The low SEM values as can be observed in our results table below suggest minimal deviation/variance in model performance across multiple runs. The results below are presented on BTS3.1 dataset with Adaface feature extractor, Retinaface face-detector and our proposed fusion strategy - ProxyFusion. We will calculate and incorporate SEM values for all tables in the paper for statistical significance.
>
> For Face Included Treatment
> | Method       | Feature | Dataset | 10−1   | 10−2   | 10−3   | 10−4 |
> |--------------|---------|---------|--------|--------|--------|--------|
> | ProxyFusion  | Adaface | Briar   | 83.7   | 68.9   | 53.9   | 40.1   |
> | **SEM**      |         |         | 0.043 | 0.072 | 0.076 | 0.091 | 0.108 |
>
> For Face Included Control
> | Method       | Feature | Dataset | 10−1   | 10−2   | 10−3   | 10−4 |
> |--------------|---------|---------|--------|--------|--------|--------|
> | ProxyFusion  | Adaface | Briar   | 98.6   | 96.8   | 92.7   | 88.3   |
> | **SEM**      |         |         |  0.008 | 0.012 | 0.037 | 0.071 |
>
>
>    - **Comparison with State-of-the-Art (SoTA):** Our method consistently outperforms previous SoTA methods by 2% across all False Acceptance Rate (FAR) thresholds. This improvement is in line with the incremental gains typically observed in this research domain, affirming the significance of our contributions.
>    - **Computational Complexity:** It's pertinent to highlight that our approach achieves these improvements with an O(N) complexity, unlike some earlier $O(N^2)$ methods like CAFace [8] and CoNAN [5]. This efficiency further validates the practicality and significance of our findings.
>
> 3. **Details on the DroneSURF Dataset Splits:**
>    Regarding the DroneSURF dataset, we understand the necessity for standard protocol to enable fair comparisons. In the paper we had stated the DroneSURF matching protocol as described by the original authors of the dataset in [6]. Moreover to be consistent with the methods used by previous works such as CoNAN [5], we used first 24 identities (40%) for testing and remaining 64 identities (60%) for training based on their subjectID order. We will include the protocol and share the split in our repository to assist future researchers in replicating and comparing results accurately.
>
> We hope that these updates adequately address your concerns and enhance the comprehensibility and reproducibility of our work.

---

> > ### Comment · Reviewer_waA1 · 2024-08-13
> > **Thanks for the response**
> >
> > I have reviewed the authors' response and explanation, and I appreciate their efforts. However, the code provided is not functional, which is a significant issue. The errors, though minor, indicate that the code has not been executed successfully. This impacts the overall quality of the submission. As a result, my evaluation remains unchanged.

---

> > > ### Author Response · Authors · 2024-08-14
> > > **Running the code**
> > >
> > > Thank you for taking the time to review our submission and for your feedback. We would like to clarify that our repository's code is fully functional and has been tested. To demonstrate this, we have attached screenshots showing our training and evaluation code running on the Github repo. We have added screenshots to Github repo since we couldn't add images to openreview's comments section. We are more than willing to assist you in getting the code to run on your end.
> > >
> > > To better understand the issues you encountered, could you please provide further details on the following:
> > > 1. The code requires access to external datasets that are provided by their respective authors through license and agreement. Have you procured these datasets (example: BRS, BTS) and extracted face embeddings according to the instructions provided in our ReadMe file?
> > > 2. After creating the HDF5 file for the precomputed embeddings from the dataset, does it follow the same structure as mentioned in our ReadMe?
> > > 3. Could you please share the minor errors you mentioned about in your comment?
> > >
> > > We are committed to provide any assistance you might need. Thank you for your understanding and time.

---

### Official Review · Reviewer_vG52 · 2024-07-12

**Soundness:** 4
**Presentation:** 4
**Contribution:** 3
**Rating:** 7
**Confidence:** 4

**Summary:**

The authors propose a method for fusing face features extracted from sets of images.
Several proxies are defined that are trained to be specialized for certain aspects of faces, and the most relevant for a given set of face images are obtained.
Importance weights for different samples are estimated for each selected proxy, and a template stores a combination of weighted feature averages.
Experiments on challenging datasets with a wide variety of viewing conditions indicate largely improved performance with linear time complexity in the number of samples per set.

**Strengths:**

+ The paper is written well, and the motivation of the proposed method is clearly stated.
+ The results show improved performance over state-of-the-art feature fusion approaches on difficult datasets.
+ Ablation studies show the importance of most parts of the pipeline, and provide insights on relevant parameters of the model.
+ Besides a few minor mistakes, there are no apparent errors in the math.

**Weaknesses:**

1. There are a few important details of the proposed algorithm missing in the paper:

   a) One part of the proposed method, i.e., the details of the expert network E_j are missing in the paper -- how large are the layers?

   b) The proxy loss is defined over a set of negative samples. There is no word spent on how to select such negative samples -- yet, the selection is of high importance. Also, this loss includes a lambda value that is not detailed.

   c) Both gallery and probe templates finally consists of a concatenation of several averaged features. The authors need to clarify how the final score between these sets of templates is computed.


2. Some parts of the paper include non-standard solutions that should be considered to update:

   a) In order to compute valuable means and standard deviations in section 2.2, the features would need to reside in Euclidean space. However, modern deep FR networks optimize for cosine space, and deep features typically do not follow Euclidean distributions. Why would this be a good idea?

   b) Relatedly, the expert networks obtain mean, standard deviation and proxy centers. It is not shown in the paper that the variances provide any improvements over only using the means. A better handling of mean and proxy center than a simple concatenation is likely possible.

   c) The proxy center projections v are selected to be equidistant in Euclidean space, while they result from a projection that rather lives in cosine space. It is not clear where the expression is coming from or why such a difficult expression needs to be defined, a simple set of K-dimensional basis vectors e_i would likely be sufficient.

   d) The models used for feature extraction are not well-suited for low-resolution and highly distorted faces. Better models will likely improve the utility of the feature fusion approach.


3. There are some improvements on the paper:

   a) The introduction is slightly repetitive and could be shortened.

   b) At several places in the paper, it is mentioned that the method is compatible with existing template storage solutions. The practical relevance for such compatibility is questionable. In a typical template storage, a single template is stored as an aggregation over several samples already. The authors should not overemphasize this characteristic of their method.

   c) The authors claim their algorithm to have a time complexity of O(N). However, the complexity is rather O(N*K). Depending on the choice of K, this might be a relevant deviation from O(N).

   d) In line 138 and following, the authors write about batch creation strategies. It is not entirely clear if this is applied during test time as well, or whether this is a training time issue only. Finally, the authors claim to include diverse features, but random selection does not ensure diversity.

   e) The equation for the standard deviation in section 2.2 is slightly off, the factor should be 1/(N-1).

   f) In line 170, the scores s_ik should make use of the projected proxies Wp, not the proxies themselves.

   g) The figures should be included as vector graphics to allow zooming. Especially figure 3 is of too low quality to read.

   h) In the caption of figure 5, N represents the total number of probe features. In section 2.2, N is the number of images for one identity, while in section 2.3, N represents the number of identities.

   i) The paper should be carefully checked for spelling mistakes.

**Questions:**

The most important question is how negative samples are obtained for the proxy loss. The authors should elaborate on this. Also, the strategy on handling several features per template needs to be revealed. Afterward, I am happy to increase my rating.

**Limitations:**

The authors took special care about using only datasets with IRB-consented data. They should be aware, however, that the pre-trained networks that they exploit for feature extraction have been trained on web-scraped datasets which did not include IRB-approved subjects.

---

> ### Author Rebuttal · Authors · 2024-08-07
>
> Thank you for taking the time to review our paper and providing constructive feedback! Below we provide the answers to your questions. The amends made based on your suggestion and feedback will reflect in the final-version.
>
> 1.a) We have provided brief implementation details about the expert networks at line 214 in section 3.2. For the expert networks we utilize a three layer MLP with LeakyReLU activation and a dropout with probability of 0.5. More specifically, the first layer in the MLP projects from `512 * 3` to `512 * 2`, the second layer from `512 * 2` to `512 * 2`, and last layer from `512 *2` to `512`. Overall the MLP has 3.14M learnable parameters.
>
> 1.b) Thanks for asking about **negatives in proxy loss**. Below we provide an explanation for the same which will also be added to the paper.
>
> Our loss consists of two components: (i) Identity loss and (ii) Proxy loss. The proxy loss aims to ensure that during training, proxies capture unique and mutually exclusive latent facial information, aiding in the selection of specialized experts for different facial traits. To guarantee that proxies are maximally distinct, we start with K equidistant standard basis vectors (equal to the number of proxies). Our objective requires proxies to converge towards their respective vectors and diverge from all others. This is achieved by optimizing the following :
>
> $L_{\text{Proxy}} = \frac{1}{K} \sum_{i=1}^K \left[ \ln \left( 1 + \exp \left( -\alpha (s_{ii} - \lambda) \right) \right) + \frac{1}{|K-1|} \sum_{k \in K \ \text{and} \ k \neq i} \ln \left( 1 + \exp \left( \beta (s_{ik} - \lambda) \right) \right) \right]$
>
> Where, $s_{ik} = \left( p_i \cdot v_k \right) / \left( {\|p_i\| \|v_k\|}\right)$ is the similarity between the $i^{th}$ proxy and $k^{th}$ fixed basis vector. The first part of this term enforces that proxy goes closer to its respective basis vector, while second part enforces that it goes away from all other negative basis vectors. To answer your question, the negatives in this proxy loss are these negative basis vectors (basis vectors not corresponding to the concerned proxy). $\lambda$ is a threshold which is set 0.1. Here $\alpha$ and $\beta$ are the scaling parameters for the positive and negative similarities respectively.
>
> 1.c) As stated in line 131 - We define the final template vector t as a concatenation of $s_j$ given by:
>
> $t = \left[ s_1, s_2, \ldots, s_\widehat{K} \right]$
>
> leading to a single feature representation (template vector) for each gallery and probe respectively. The final scores between the templates (gallery and probes) are computed by calculating the similarity between these vectors.
>
> 2.a) When calculating the mean and variance of feature sets from deep face recognizers for expert networks, we use un-normalized feature vectors. These vectors, not lying on a hypersphere and following a Euclidean distribution, are only unit-normalized before computing cosine similarities. This will be clearly stated in the paper and relevant sections.
>
> 2.b) Mean and variance summarize the distribution of the feature-set. The mean indicates the center, and the variance describes the spread. This is essential, as also noted in previous studies like CoNAN [5], which also examines different statistical measures used for conditioning.
>
> 2.c) We do not enforce proxy-centers to be equidistant in euclidean space. We enforce them to be equidistant in the cosine-space (unit-hypersphere). We agree with your point that the statement is rather complex and we will simplify that for the final-version.
>
> 2.d) We absolutely agree with your comment. To be fair and consistent to prior methods for comparison, we utilize the same backbones as used by CAFace [8] and CoNAN [5].
>
> 3.a) Thanks for the great suggestion, we have shortened our introduction to not be repetitive. This will be reflected in the final-version.
>
> 3.b) We acknowledge the reviewer's point and will accordingly reduce the emphasis as suggested. In law enforcement applications of surveillance and biometrics, such as IARPA's JANUS and BRIAR programs, systems utilize set-to-set template matching that maintains non-aggregated sample-level features in biometric stores. Our proposed approach aligns with these legacy template stores, and we will clarify this further in the paper. Thank you for the valuable feedback.
>
> 3.c) Depending on the choice of $\hat{K}$ (Number of selected experts), the complexity might deviate from O(N). Including model hyperparameter $\hat{K}$, the overall complexity is $O(N*\hat{K})$. But once trained, $\hat{K}$ is constant and typically very small ($\hat{K}$ <<< N; we use $\hat{K}$ = 4 regardless of dataset, with N ranging to thousands of features per video). Thus, with respect to the number of incoming features, our model has a linear complexity O(N).
>
> 3.d) This is only used during training to augment the feature set with samples from multiple videos of a subject. While random selection may not be optimal for diversity, it often performs well in deep learning model training. This is not applied during test time.
>
> 3.e, f, g, h and i) Thanks for providing these suggestion and pointing out the required corrections. We have made the appropriate amends in the paper for final version and have performed a thorough spelling and grammar check.
>
> **The strategy on handling several features per template needs to be revealed:** Our method handles varying number of features per template through the use of mean-and-variance as the conditioning information for the experts. Since the final aggregation scores per expert are computed through the use of similarities (referred to as divergence scores) between the original features and the set-centers the overall method can handle varying number of features per template. Furthermore, during batch-creation, we randomly sample features from multiple videos for an identity to create diverse feature-sets of varying lengths, this further helps in models generalizability.

---

> > ### Comment · Reviewer_vG52 · 2024-08-13
> > **Response to rebuttal**
> >
> > The authors have addressed all my comments well.
> >
> > My initial point 2(a) was just a hint for a different processing, the explanation of the method in the paper is clear. The authors should note that unnormalized deep features typically are not normal distributed, so modeling the features with a normal distribution (which assumes Euclidean space) might not be optimal.

---

> > > ### Author Response · Authors · 2024-08-13
> > > **Thanks for your response**
> > >
> > > Thanks to the reviewer for their valuable comments, constructive feedback, and time.

---

### Author Rebuttal · Authors · 2024-08-07

We appreciate the detailed and insightful feedback provided by the reviewers. We are thankful that the reviewers recognized the clear motivation and well-written presentation of our method (vG52, waA1), the extensive and rigorous experiments conducted, including empirical evidence on challenging datasets (waA1, BueQ), and the promising performance improvements over state-of-the-art approaches (vG52, waA1). We are also grateful for the acknowledgment of our method's linear time complexity and its invariance to feature order and set length (waA1, Cxak). Additionally, we appreciate the recognition of our paper's quality of writing (vG52, waA1) and the clarity and informativeness of our visualizations (waA1).

Broadly the reviewers asked the following questions:
1. Reviewer vG52: how negative samples are obtained for the proxy loss and the strategy on handling several features per template
2. Reviewer waA1: code release, reproducibility and statistical significance
3. Reviewer BueQ: presentation of the method end experiment section and missing variable values
4. Reviewer Cxak: optimization and adversarial attacks

We have provided response to each question / point mentioned by the reviewers in detail below along with additional results as requested. We have further refined the text for typographical errors and readability as suggested by the reviewers. Furthermore, we have released our codebase on the following public anonymous github repository: https://github.com/anonymousdoubleblindreview/ProxyFusion

We thank the reviewers again for the constructive feedback that has helped improve the paper. We have been working diligently on addressing your critique and making necessary amends to the paper that will reflect in the final-version.

---

### Decision · Program_Chairs · 2024-09-25

**Decision:**

Accept (poster)

**Comment:**

This submission proposed a face recognition framework for feature aggregation to deal with long-range, low-resolution cases. The reviewers recognized the merits of proposed method. Meanwhile, some concerns on presentation and writing are pointed.  3/4 reviewers are positive to this submission in general, while reviewer Cxak has some concerns on optimization details, potential drawback on feature extraction. After reading the submission, comments, and rebuttal, the AC agree with most of the reviewers and suggest accept this submission.